# Potential for Use of Species in the Subfamily *Erynioideae* for Biological Control and Biotechnology

**DOI:** 10.3390/microorganisms12010168

**Published:** 2024-01-14

**Authors:** Andrii P. Gryganskyi, Ann E. Hajek, Nataliya Voloshchuk, Alexander Idnurm, Jørgen Eilenberg, Romina G. Manfrino, Kathryn E. Bushley, Liudmyla Kava, Vira B. Kutovenko, Felicia Anike, Yong Nie

**Affiliations:** 1Division of Biological & Nanoscale Technologies, UES, Inc., Dayton, OH 45432, USA; 2Department of Entomology, Cornell University, Ithaca, NY 14853, USA; aeh4@cornell.edu; 3Faculty of Plant Protection, Biotechnology and Ecology, National University of Life & Environmental Sciences of Ukraine, 03041 Kyiv, Ukraine; voloshchuk_m_nataliia@ukr.net (N.V.); kavalyuda@ukr.net (L.K.); 4Department of Food Science, Pennsylvania State University, University Park, PA 16802, USA; 5School of BioSciences, University of Melbourne, Parkville, VIC 3010, Australia; alexander.idnurm@unimelb.edu.au; 6Department of Plant & Environmental Sciences, University of Copenhagen, DK-1870 Frederiksberg, Denmark; jei@plen.ku.dk; 7CEPAVE—Center for Parasitological & Vector Studies, CONICET-National Scientific & Technical Research Council, UNLP-National University of La Plata, La Plata 1900, Buenos Aires, Argentina; manfrino@cepave.edu.ar; 8USDA-ARS Emerging Pests & Pathogens Unit, Ithaca, NY 14853, USA; keb45@cornell.edu; 9Agrobiological Faculty of Plant Protection, National University of Life & Environmental Sciences of Ukraine, 03041 Kyiv, Ukraine; virakutovenko@gmail.com; 10Department of Natural Resources & Environmental Design, North Carolina Agricultural & Technical State University, Greensboro, NC 27401, USA; felicia.anike@gmail.com; 11School of Civil Engineering & Architecture, Anhui University of Technology, Ma’anshan 243002, China; nieyong@ahut.edu.cn

**Keywords:** insect biocontrol, cultivability, genomics, entomopathogens, geographic distribution, host range

## Abstract

The fungal order *Entomophthorales* in the *Zoopagomycota* includes many fungal pathogens of arthropods. This review explores six genera in the subfamily *Erynioideae* within the family *Entomophthoraceae*, namely, *Erynia*, *Furia*, *Orthomyces*, *Pandora*, *Strongwellsea*, and *Zoophthora*. This is the largest subfamily in the *Entomophthorales*, including 126 described species. The species diversity, global distribution, and host range of this subfamily are summarized. Relatively few taxa are geographically widespread, and few have broad host ranges, which contrasts with many species with single reports from one location and one host species. The insect orders infected by the greatest numbers of species are the *Diptera* and *Hemiptera*. Across the subfamily, relatively few species have been cultivated in vitro, and those that have require more specialized media than many other fungi. Given their potential to attack arthropods and their position in the fungal evolutionary tree, we discuss which species might be adopted for biological control purposes or biotechnological innovations. Current challenges in the implementation of these species in biotechnology include the limited ability or difficulty in culturing many in vitro, a correlated paucity of genomic resources, and considerations regarding the host ranges of different species.

## 1. Introduction

The fungal order *Entomophthorales* in the *Zoopagomycotina* includes at least 246 species of arthropod pathogens [1], many of which are well known for their ability to cause epizootics and change the behavior of infected hosts [2]. Their role in biocenoses is extremely important because they can function as regulators of arthropod populations and thus play a role in ecosystem homeostasis. Within the *Entomophthorales*, the largest family is the *Entomophthoraceae*, which consists exclusively of arthropod pathogens. This family was divided into two subfamilies in 2005, the *Entomophthoroideae* and the *Erynioideae* [3]. The *Erynioideae* is the larger of these two subfamilies, containing six genera, namely *Erynia*, *Furia*, *Orthomyces*, *Pandora*, *Strongwellsea*, and *Zoophthora*. Fungi in these genera have diverse ecological, physiological, and morphological adaptations (Figure 1) and evolved to infect a wide range of arthropod species using their ballistic conidia. Their hosts inhabit diverse ecosystems, including agriculture and forestry. In particular, *Erynioideae* infect insects that are recognized as pests of various important crops worldwide [4,5,6]. In addition to attacking arthropod pests directly damaging crops and forests, some of the arthropod hosts of species in this subfamily include vectors of diseases that impact humans, livestock, and crops.

Due to the observed capability of *Entomophthorales* to cause massive mortality of insect hosts, questions about their potential use for biocontrol and various biotechnological applications are often raised. Exotic strains and species have been released for classical biocontrol of diverse insects in many countries, with some successful establishments and pest control [7]. However, presently, no species of Entomophthorales are commercially available as biopesticides. Species within this group differ from one another in numerous ways that impact their potential development as biopesticides [8]. One of the most important attributes to consider in this regard is the cultivability of these fungal species in vitro, which plays a critical factor in their propagation for potential application as biological control agents. This feature can be expressed to various degrees: from total inability to isolate fungal strains in vitro to routine transfers of the isolates and preservation of their cultures in culture collections to research on improving in vitro growth toward potential mass production. Additionally, for successful biological control, the host range of the pathogens must be known and is crucial in both identifying suitable fungi for specific target pests as well as in avoiding potential impacts on non-target arthropods. Furthermore, the natural habitats of these fungi and their geographical distributions are important for consideration of development for biological control as regions around the world differ in the regulation of species to be used for pest control that are native vs. non-native [9]. In addition, knowledge of the habitats where these fungi are naturally active will provide information about their ecological adaptability and long-term survival under diverse conditions.

The primary objective of this research was to characterize several critical aspects of the lifestyles of the species within the six genera in the *Erynioideae*, here referred to by the acronym EFOPSZ. An overall consideration of the species in this group has not previously been undertaken. We have analyzed these vital characteristics for species of the *Erynioideae*, and we aim to pinpoint which species demonstrate the most potential for future use in biological control. Moreover, we are keen on identifying the specific insect groups for which the application of these species as biological control agents might be most successful. We predict that the assembly of this new information could be important for potential applications of this group in various aspects of biotechnology, particularly in the development of biological control agents for pest management.

## 2. Materials and Methods

### 2.1. Literature Analysis

We sought all literature related to the six genera in the *Erynioideae* through the use of the Web of Science, Scopus, and Google Scholar, examining publications since 1888 [10] using keywords and names of genera included in this study. We also examined the information associated with the species and strains deposited in the U.S. Department of Agriculture, Agricultural Research Service Collection of Entomopathogenic Fungal Cultures (ARSEF, Ithaca, NY, USA); the American Type Culture Collection (ATCC, Manassas, VA, USA); and CBS-Westerdijk Institute KNAW Fungal Biodiversity Centre, also known as Central Bureau of Fungal Cultures (Utrecht, Netherlands). The traits that were investigated are geographical distribution, host range, type of habitat, and documented ability to grow in vitro.

### 2.2. Distribution Map

A map of the number of recorded species was created using StepMap GmbH software (Berlin, Germany). Countries and regions were colored according to the number of recorded species from lightest (less than 5 species) to darkest (over 50 species described) pink. Green indicates none have been reported. We consider the species as (1) local if the distribution range covers only one continent, (2) cosmopolitan or broadly distributed with records on at least two continents, and (3) ubiquitous if distribution records cover three or more continents.

### 2.3. Phylogenetic Tree

To generate a dataset of EFOPSZ taxa, we downloaded 18S and 28S sequences of identified species with accurate nomenclature from GenBank. All sequences were initially aligned, their alignments were manually adjusted, and ambiguous regions were excluded from the alignments using Mesquite 3.04 version [11]. Phylogenetic relationships were determined by the neighbor-joining (NJ) algorithm, and the tree was visualized in PAUP* 4.0 [12].

## 3. Results

### 3.1. Geographic Distribution

Species in the EFOPSZ group have been recorded from all continents except Antarctica. However, the number of described species differs significantly between countries and regions (Figure 2). The most records are from several Central European countries (especially Switzerland and Poland) and the United Kingdom. Many species have also been found in North America (USA and Canada), other European countries, and China. Only a handful of reports are from South America and across a considerable area of Asia and Oceania. The continent least documented for these fungi is Africa, where EFOPSZ fungi have been reported from only four countries. They are also sparsely reported in countries in South America other than Argentina, Brazil, and Chile and are reported from a few countries in the Middle East. However, the map in Figure 2 most likely does not represent the actual species distributions but rather the situation regarding our knowledge of this fungal group in particular countries where more sampling has occurred. It is obvious that climatic conditions in many countries of Africa or South America might be very favorable for species in the *Erynioideae*, but little research has yet been undertaken to describe *Erynioideae* and their host ranges from these regions.

There were clear differences in patterns of species distribution among the six genera. Each genus contains both ubiquitous and cosmopolitan species as well as local ones, and they are not grouped in any specific way on the phylogenetic tree (Figure 2). It is very possible that many of the species of these genera that we classify as local have much broader distributions but have not been sampled broadly. We consider the broad distributions of many species as advantageous for future biocontrol agents since this feature might indicate a significant level of adaptability and ability to survive the environmental conditions in different climatic environments due to their ecology, pathogenesis, and specialization [13]. 

Another perspective on geographical distribution can be obtained from analyzing culture collection deposits. The largest insect pathogen collection in the world is the USDA/ARS entomopathogenic fungi collection (ARSEF), which includes almost 15,000 occurrence records [14]. Although ARSEF has deposits from all over the world, most samples are from the USA and then from Europe. We hypothesize that isolates from the rest of the world are less represented due to a lack of sampling. Despite the scarcity of well-recorded data, at least 53 species out of the 125 valid EFOPSZ species might be considered as cosmopolitan, and at least 25 as ubiquitous (Figure 3, Table 1). Many of these might become ubiquitous due to the worldwide nature of human agricultural activities, which spread many crops worldwide along with their pests. One of the best examples of a human-mediated distribution might be *Pandora gloeospora*, found on several continents in mushroom-growing farms [15]. 

### 3.2. Host Specificity

EFOPSZ fungi show a range of host-specificity. One-third of EFOPSZ parasitize two or more insect families. The species *Erynia conica*, *E. rhizospora*, *E. selpulchralis*, *P. batallata*, *P. blunckii*, *P. echinospora*, *P. nouryi*, *Z. aphidis*, *Z. canadensis* are pathogenic to representatives of at least two families. An absolute generalist is *Z. radicans*, which infects insects in 7 orders and 21 families (Table 1). However, two-thirds of EFOPSZ fungal species show some host-specificity and can infect only a narrower range of insects, usually attacking members of the same genus or family. 

The flies (order *Diptera*) are the most frequent hosts for EFOPSZ fungi, as more than one-third of these fungal species were found killing *Diptera*. Nearly 25 percent of EFOPSZ fungi infect insects in the order *Hemiptera* (31 pathogenic species), and nearly half that number were found infecting *Coleoptera* (16 pathogens) and *Lepidoptera* (15 pathogens). Within the *Diptera*, families most attacked by EFOPSZ fungi are *Calliphoridae* (eight pathogen species); *Tipulidae* (seven); *Muscidae*, *Psychodidae*, and *Sciaridae* (six each); and *Chironomidae* (five). In the genus *Strongwellsea*, species specialize exclusively in four dipteran families: *Anthomyiidae*, *Muscidae*, *Sarcophagidae*, and *Scatophagidae*. Among *Hemiptera*, the families *Aphididae* (14), *Miridae* (7), and *Cicadellidae* (6) are most infected by EFOPSZ fungi. All other insect families have less than five pathogenic EFOPSZ species infecting them (Figure 4). 

### 3.3. Biological and Ecological Characteristics of EFOPSZ Fungi as Biocenose Components

Most species in the *Erynioideae* primarily infect insects in natural and agricultural environments. These habitats include aquatic biocenoses, forests and natural areas, and agrocenoses. It might be more precise to discuss the distribution of insect hosts, even if fungal infections can lead to infected insects relocating from their typical habitats [86]. Many EFOPSZ species infect the imago (adult stage) of hosts. Among holometabolous hosts, species in the genus *Strongwellsea* only infect adults, while some species from the other genera infect larvae or nymphs. No EFOPSZ species have been found attacking insect eggs. 

These different host life cycle stages may occur in various ecosystems, so this factor should also be considered. In most cases, infected insects are found and collected on plant parts, partly due to the so-called climbing effect caused by many EFOPSZ fungi [166]. This altered behavior may also be attributed to better visibility for researchers compared to the soil surface beneath vegetation. A comprehensive analysis of the distribution of entomophthoralean fungi in European biocenoses, focusing on forests and agrocenoses, was carried out by Bałazy [17]. Bałazy’s analysis emphasizes the importance of insect mobility, particularly because many insect hosts have wings and can migrate to neighboring ecosystems, spreading infection. This mobility is supported by the collection of dispersing aphids infected with *P. neoaphidis* [167] and the isolation of *P. delphacis* from planthoppers caught on a weather ship off the coast of Japan [76]. 

Most EFOPSZ fungi are prevalent in aboveground ecosystems. The cadavers of insects infected with EFOPSZ are often found on wild and cultured plants in various ecosystems. The presence of representatives of the genus *Zoophthora*, in particular, is well-documented in numerous agricultural crops, orchards, and different types of forests. Species like *P. dipterigena*, *P. philonthi*, *Z. anglica*, *Z. miridis*, *Z. opomyzae*, *Z. petchii*, *Z. phytonomi*, and *Z. radicans* are commonly observed in annual and perennial crops, meadows, pastures, orchards, and forests. These species seem to be well-adapted to drier habitats. Species of the genera *Strongwellsea* and *Zoophthora* appear to be the most adaptable to a wide range of habitats, whether natural or human created. Furthermore, aphid pathogenic species, like *P. neoaphidis* and *P. nouryi*, have been found worldwide in many crops and are commonly observed at different temperatures and humidities.

All *Entomophthorales* require high humidity to release and disperse their conidia [168]. Interestingly, half of the species in the genus *Erynia* were found in aquatic or notably moist areas, e.g., *E. aquatica*, *E. conica*, *E. curvispora*, *E. nematoceris*, *E. ovispora*, *E. sepulchralis*, and *E. variabilis*. These species may serve as efficient biological control agents for insects requiring aquatic habitats during specific life stages due to their higher humidity needs compared to other species in this group. Additionally, five species in the genus *Pandora*, one species in *Furia*, and one species in *Zoophthora* were found in moist habitats. However, no *Strongwellsea* species were recorded in explicitly aquatic or moist environments (Table 1). 

Soil is an unusual habitat for predominantly insect-pathogenic EFOPSZ. Nevertheless, at least one species, *Pandora nouryi*, infects root aphids (*Pemphigus*) and follows its hosts to this habitat, becoming a soil dweller [78]. *Zoophthora myrmecophaga* infects ants that move along their paths on the soil surface. *Pandora brahminae*, which infects scarabs inhabiting the soil surface, also might be considered soil inhabitants. 

### 3.4. Cultivability

Few species in the EFOPSZ group have been isolated into pure culture or even had their cultivability tested. Most species have been described only from insect cadavers, and there are no cultures preserved. The USDA ARSEF culture collection contains fungal strains isolated from infected insects. While most strains belong to the *Ascomycota*, entomophthoralean fungi are also well represented. This collection preserves 683 total isolates of EFOPSZ [14]. These include 28 species known in the genera *Erynia*, *Furia*, *Pandora*, *Strongwellsea*, and *Zoophthora*, as well as 36 isolates from these 5 genera, which are not yet identified at the species level. Most species are represented by a single or just a few isolates. However, there are over a hundred isolates representing species such as *P. neoaphidis* and *Z. radicans*, which reflects the common occurrence and easy cultivability of those species. 

Few EFOPSZ fungi can be cultivated on typical fungal nutritional media such as malt extract or potato dextrose agar or in the corresponding liquid media [168]. In the past, to ensure the fungal growth of entomopathogens, special media containing animal protein from additives such as liver, extracts of fresh or dried insects, blood serum, or egg yolk were used [23,169], providing the pathogens with specific nutrients absent in the usual laboratory media. Sometimes rare and exotic media components such as fly fat bodies are used to stimulate spore germination or hyphal growth. The addition of yeast extract, arginine, or other amino acids to the medium substantially improves the growth of entomophthoralean fungi. Nowadays, the most used liquid medium for entomophthoralean growth is Grace′s Insect Medium (Sigma-Aldrich, St. Louis, MO, USA), often with additives such as fetal bovine serum (ThermoFisher Scientific, Waltham, MA, USA). 

Larger-scale production using simpler media has been developed for a few species. A method for producing *Z. radicans* dry-formulated mycelium has been developed with sporulation equivalent to cadavers. Recent examples of successful production on a large laboratory scale of the fungus *P. cacopsyllae* are the studies by Muskat et al. ([170,171], in press). This fungus that infects psyllids of the genus *Cacopsylla* has been fermented, encapsulated, and tested for above-ground application.

The most fastidious EFOPSZ species can so far only be cultured in vivo. They require the maintenance of an insect colony either in the laboratory or in the natural environment to maintain their population and complete their life cycle. The best results are obtained from using the natural hosts of the pathogenic fungus. In vivo production is the most difficult and labor-intensive method for growing entomopathogenic fungi.

The ability to grow EFOPSZ in culture is often connected with the ability of these fungi to infect insects at various stages of their life cycle, or at least stages other than the imago (adult), as seen in species like *E. curvispora*, *Z. bialovienzenzis*, *Z. lanceolata*, *Z. phytonomi*, and *Z. radicans* [17,23]. One of the remarkable characteristics of EFOPSZ fungi, particularly those with high-host or life-stage specificity, is the loss of their vigor and viability after several transfers on laboratory media despite strong initial growth [168,172]. These features of highly host-specific members of the EFOPSZ can pose a challenge to mass production in biotechnology.

### 3.5. Molecular Data, Genomics, and Biotechnology

Molecular data have been obtained for only ca. one-fifth of *Erynioideae* species, which has complicated their identification. Most available data used for the molecular taxonomy of this fungal group are the small (18S) and large (28S) RNA subunit sequences, which are successfully used to build phylogenetic trees and provide molecular identification of species (e.g., see Figure 2). The most frequently used primers are LROR and LR3 or LR5 (28S) and NSSU1088r and NS24 (18S). Amplification of the internal subscribed spacer (ITS) region, which is usually used as a barcoding gene for fungal species identification, can be challenging in the case of the *Erynioideae*, possibly because of its length, which varies from 0.9 to 1.3 k nucleotides in known species using the typical fungal barcode primers ITS1f and ITS4. However, partial ITS sequences (ITS2 region) were used to genotype the species of *Strongwellsea* and describe several new species of this genus [128,129,131,132]. *Pandora formicae*, *P. gammae*, *P. kondoiensis*, *P. neoaphidis*, *P. nouryi*, and *Zoophthora radicans* have also been genotyped with ITS sequences [86,173,174,175,176]. Some species have information available for genes encoding elongation factor 1-alpha, RNA polymerase II subunits (*RPB1* and *RPB2*), mitochondrial small subunit ribosomal RNA, white collar-1 protein, beta-tubulin (*btub*), elongation factor 1 alpha-like protein (*efl*), cell division control protein 25 (*CDC25*), chitin synthase (*CHS3*), chitin deacetylase (*CHD1*), chitinase 1 (*CHI1*), endochitinase (*CHT1*), triacylglycerol lipase (*LIP2*), glucan binding protein (*GBP1*), subtilisin-like protease precursor (*SPR2*), polyketide synthase (*PKS*), triacylglycerol lipase precursor (*LIP1*), trypsin-like serine protease precursor (*TRY1*). The most intensively genotyped species are definitely *P. neoaphidis* and *Z. radicans*, from which most of the aforementioned gene fragments have been obtained [174,175,177,178,179,180,181,182,183]; these species have also been considered the most promising agents for biocontrol. *Zoophthora radicans* transcriptomes were also extensively investigated in regard to fungal pathogenicity [184]. As an example of limited genetic information, no DNA sequence information is available from the single species in the genus *Orthomyces.*

In addition to using these species directly for biological control, other potential biotechnology applications based on the use of genes or proteins derived from these fungi have yet to be explored. The early diverging lineages of fungi, here defined as the paraphyletic group, not including the *Dikarya* (e.g., *Ascomycota* and *Basidiomycota*), have emerged through the characterization of their genomes as distinct among fungi in containing numerous genes and traits shared with animals that were lost in more derived members of the *Dikarya*. Such homologs may, in the future, provide new insights into fundamental biology or even lead to therapeutics for human health. Entomopathogenic fungi themselves are well known to produce diverse small-molecule secondary metabolites/natural products with activity against insects [185]. Genome sequences have revealed evidence of greater biosynthetic capability for such molecules, even among some lineages of early diverging fungi [186], and the potential for EFOPSZ fungi has not been thoroughly explored. Additionally, entomopathogens, including some in *Entomophthorales*, also produce protein toxins that show insecticidal activity [187,188].

At present, just a single species from the *Erynioideae* has had its genome sequenced, i.e., *Z. radicans* [189]; this was possible in part due to the ease of its cultivability. The genome was generated as part of a large collection of species in a study that addressed the ploidy levels of the early fungal lineages, and so features about what this genome contains have not been described yet in detail. This genome sequence carries the information for how *Z. radicans* completes its lifecycle, including as an entomopathogen, but the mechanism is not immediately obvious. The *Z. radicans* genome is large, with the assembly at over 650 Mb currently divided over nearly 7000 scaffolds and estimated to encode over 14,000 genes. The large size is due to the large amount of repetitive DNA within the genome (Figure 5). The entomopathogens in the *Hypocreales* (*Ascomycota*) contain numerous gene clusters for the synthesis of secondary metabolites, one of the best known being the cluster for the immunosuppressive cyclosporin from *Tolypocladium inflatum* [190], as well as other types of toxins (e.g., enterotoxins) with possible roles in altering host behavior [191]. However, such detailed information based on the *Z. radicans* genome is not yet available, and a cursory examination of the genome indicates no examples of gene clusters for the synthesis of toxins.

As pointed out above, one challenge toward generating more genome sequences is being able to obtain sufficient DNA, usually through culturing of isolates in vitro, which has not been possible for many of these species. Difficulties with culturing EFOPSZ fungi make sequencing their genomes complicated because of the challenge of isolating high-quality and high-molecular-weight DNA and RNA for sequencing. For many species, extraction of total DNA from the host insect cadaver might be the only option. However, the recent advances in single-cell genomics [192] may provide a way in the near future to generate more genomic information about the genetic composition and potential virulence factors of EFOPSZ species. Once identified, these genetic components could be utilized in vector-based expression systems for application as biopesticides. There are also a few genomes available for the closely related subfamily—the *Entomophthoroideae*—including *Entomophthora muscae*, *Entomophaga maimaiga*, *Massospora cicadina*, as well as other species in *Entomophthorales*: *Conidiobolus coronatus*, *Neoconidiobolus thromboides*, and *Basidiobolus meristosporus* [193]. Summarizing genome features for the available entomophthoralean genomes, it can be predicted that the genomes of most EFOPSZ fungi are much larger compared to the average ascomycete fungal genome size (40–60 kB) and can reach 600,000–1,000,000 kB in size and consist of many duplicated gene copies and repeated regions [194].

**Figure 5 microorganisms-12-00168-f005:**
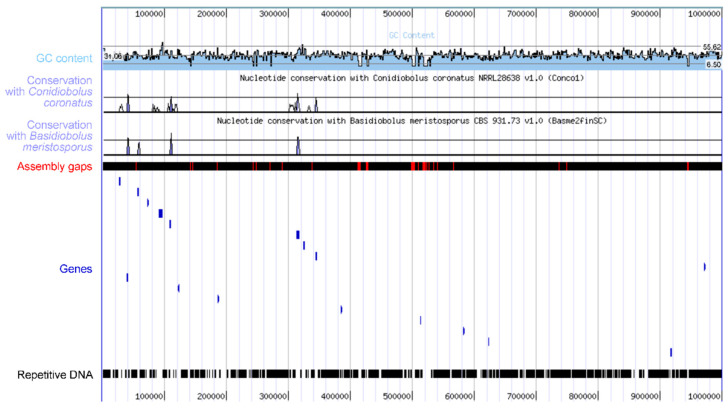
The *Z. radicans* genome features large amounts of repetitive DNA between coding regions. This figure is a modified version of the MycoCosm [195] visualization of 1 Mb of sequence on contig 1, showing in blue only 17 genes across the region, separated by repetitive DNA in black. Conservation of DNA sequences with two other sequenced species in the *Entomophthorales* is low.

## 4. Discussion

The goal of using entomopathogenic fungi in various biotechnological applications, particularly to control populations of agricultural insect pests, has existed for several decades. However, entomophthoralean species have not been successfully developed and applied for biological control despite numerous attempts. The major challenges in their application as biocontrol agents include the difficulty with the cultivation of many species, requirements for specific abiotic conditions in the field, and the potentially low survival rates of these fungi outside of the host. However, EFOPSZ fungi possess significant potential, which is still largely unexplored. Recent advances in genome sequencing technologies may allow researchers to access key genetic factors that are involved in virulence against insect hosts, even in those EFOPSZ species that cannot be easily cultivated, and biotechnology could then potentially be used to deploy these in various ways for pest control.

The ability of entomophthoralean species to infect insects from different families or even from different orders increases the diversity of target insect species for developing efficient biocontrol measures and selection of suitable pests to control. At the same time, the broad host ranges of some of these fungi make use of newly developed biological control remedies riskier as they may also affect non-target insects, including those that are beneficial for natural ecosystems and humans. 

While further research investigating host range among EFOPSZ is needed, the search for potential biocontrol agents within the *Erynioideae* might use information on current host ranges and distributions, targeting hosts belonging to insect taxonomic groups that are known to be attacked by EFOPSZ (Table 1). Even more important, prediction of possible non-targets of entomopathogenic species should account for insects within those same taxonomic groups in addition to pollinators and other beneficials. To some extent, it is an advantage if a species already has been found on several continents and thus might be adapted and developed for use in biocontrol over a larger area. These species, with high adaptability to various environments, are promising candidates for targeting widespread insect hosts. Each genus of the EFOPSZ group has several species that are distributed on at least three continents: in the genus *Erynia*, *E. aquatica*, *E. conica*, *E. ovispora*, and *E. rhizospora*; in the genus *Furia*, *F. americana*, *F. gastropachae*, *F. ithacensis*, and *F. virescens*; and in the genus *Pandora*, *P. blunckii*, *P. bullata*, *P. delphacis*, *P. dipterigena*, *P. gammae*, *P. neoaphidis*, and *P. nouryi*. Many ubiquitous species are included in the genus *Zoophthora*: *Z. aphidis*, *Z. geometralis*, *Z. occidentalis*, *Z. phalloides*, *Z. phytonomi*, and *Z. radicans*. Two species of *Strongwellsea*, *S. castrans* and *S. magna*, have been found on two continents. With increasing research on entomophthoralean fungi, we hypothesize that it is likely that a larger number of ubiquitous species will be identified.

Representatives of the genera *Erynia*, *Pandora*, and *Zoophthora* are among the important infective agents of insects under field conditions. The attempts at the introduction of EFOPSZ species on different continents have had success with *Z*. *radicans* in Australia using a strain originated from Israel to control the spotted alfalfa aphid, *Therioaphis maculata* [78]. 

A lower need for humidity could be considered an advantageous feature for potential biological control preparations using EFOPSZ species. Aquatic species like *E. aquatica* might be successfully used only in wet habitats where they are highly adapted to the moist environment, and this could restrict application compared to the species found in diverse ecosystems. The broad ecological and geographical range of *Z. radicans*, recorded from numerous agricultural and natural habitats, makes this species unique within the EFOPSZ.

Cultivability is perhaps the main factor that determines the potential success of any biotechnological application with EFOPSZ. If the fungus is hard to cultivate on artificial media, then the only way to apply it as a biocontrol agent is to keep it in vivo, infecting the insect population either in nature or under lab conditions. However, this is costly, cumbersome, and risky. Development of the biotechnological process and especially scaling it up demands easy culturing without losing virulence. The challenges of isolation into a pure culture of the majority of EFOPSZ species, along with loss of vigor during numerous culture transfers, significantly complicate the research on and development of potential biocontrol.

A problem with successful pest control with many fungi is the level of susceptibility of the active organisms to external factors, such as fluctuations in temperature, humidity, and rainfall. Climate changes may significantly impact the relationship between fungi, insects, and crops and the interactions among them [196]. Furthermore, additional information needed for the eventual production of EFOPSZ as biopesticides would be the development of optimal methods for formulation and application.

## 5. Conclusions

Analysis of the available data on virulence, growth in vivo and in vitro, formulation, and field testing suggests that one promising candidate for the development of efficient biological control agents would be the species *Z*. *radicans.* This species seems to have the potential to control a range of lepidopteran larvae in many agricultural and forest ecosystems. Another species, *P. neoaphidis*, has great potential for control of numerous aphid species in cereals and legumes. Of special importance is the worldwide distribution of aphids impacting crops and, thus, the large market that exists. In orchards, *P. cacopsyllae* has recently been proven to possess a potential for control of psyllid pests. A major objective here is that fruits are high-value crops that may favor biocontrol options. For the moist and aquatic habitats, there are several species infecting dipterans. *Erynia aquatica* and *E. conica* may have the potential for mosquito control; however, little is known about their virulence and growth in vitro. Obviously, the aforementioned factors are not the only factors determining the success or failure of biocontrol development; however, they play essential roles. However, advances in genome sequencing methods may allow researchers to access virulence factors and other genetic factors of these fungi that could be harnessed for future biotechnological solutions.

## Figures and Tables

**Figure 1 microorganisms-12-00168-f001:**
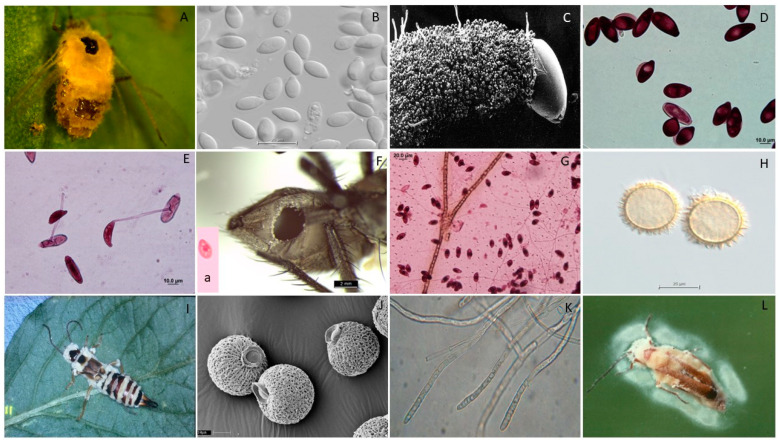
Examples showing the diversity of EFOPSZ cell types and hosts. Aphid infected with *Zoophthora radicans* (**A**). *Pandora cacopsyllae* primary conidia, typical for this genus morphology (**B**). *Pandora sciarae* conidiophores covering the body of a fungus gnat (**C**). Primary and secondary conidia of *Pandora neoaphidis* (**D**). Germination of *Z. radicans* primary conidia with secondary conidia (**E**). Cadaver of *Delia radicum* with abdominal hole where *Strongwellsea* sp. primary conidia are actively discharged (**F**); single nuclear conidium (**a**). Nuclear stain of *P. neoaphidis* primary conidia (**G**). *Strongwellsea selandia* round resting spores with spines (**H**). White layer of *Zoophthora forficulae* conidiophores penetrating whole earwigs cadaver except thick cuticular plates and limbs (**I**). Incrustation of the *Zoophthora independentia* resting spores (**J**). Septation of the *Z. radicans* vegetative mycelium (**K**). Whitish “halo” of the *Pandora lipai* conidia discharged from the whitish/yellowish conidiophore layer covered the soldier beetle (**L**).

**Figure 2 microorganisms-12-00168-f002:**
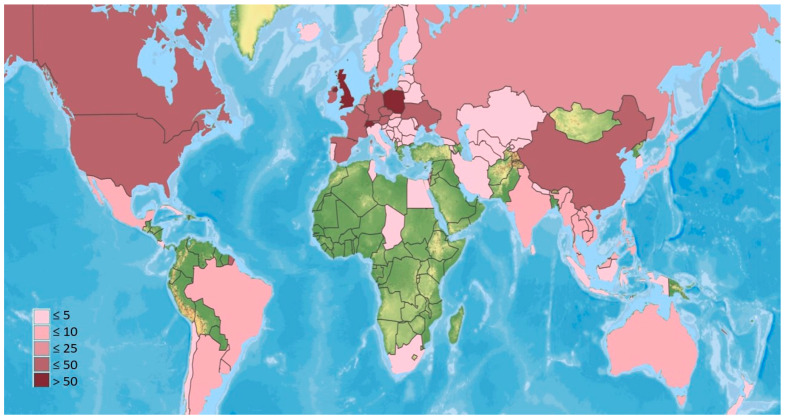
Number of EFOPSZ species recorded for different countries. Green indicates none reported.

**Figure 3 microorganisms-12-00168-f003:**
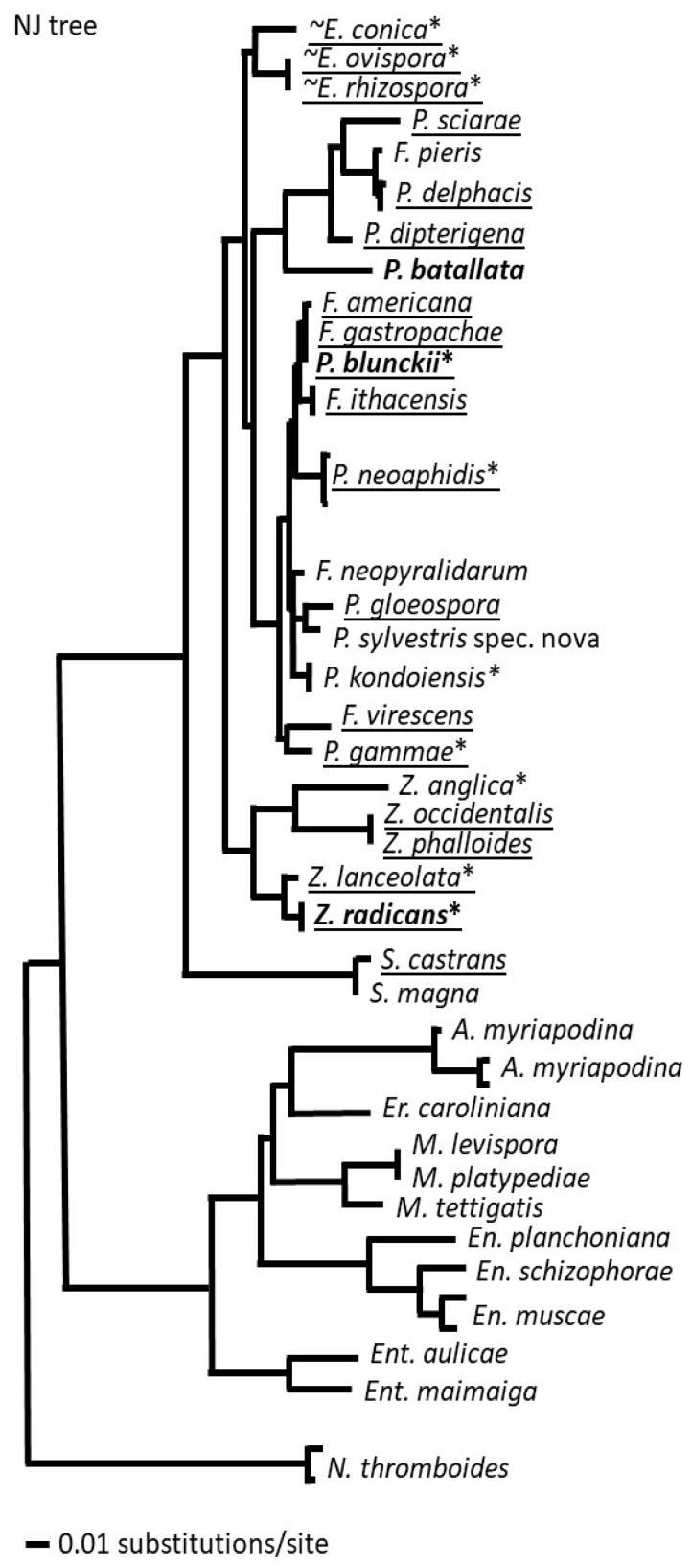
A phylogenetic tree of species in the genera *Erynia*, *Furia*, *Pandora*, *Strongwellsea*, and *Zoophthora* of the subfamily *Erynioideae*: **~**—aquatic or moist habitats, *—cultivable; generalists in bold, ubiquitous species underlined. Members of the *Entomophthoroideae* and genus *Neoconidiobolus* are provided as outgroups.

**Figure 4 microorganisms-12-00168-f004:**
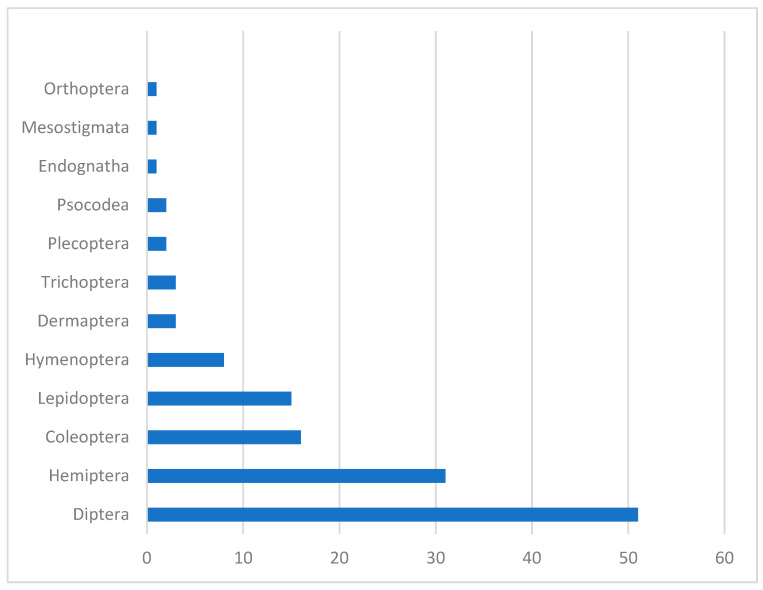
How many EFOPSZ species infect different arthropod orders.

**Table 1 microorganisms-12-00168-t001:** Geographic distributions and arthropod hosts of EFOPSZ fungi.

Species	Host Order (All *Insecta*, Except as Noted)	Host Family	Locations in Detail	Reference (Based on Literature Search and ARSEF Collection Records)
*~Erynia aquatica ** (2)	*&Diptera*	*Culicidae*	Europe: Poland, RF, Spain, Sweden, Switzerland, Ukraine; Nepal; USA	[16,17,18,19,20,21,22,23,24,25]
*~E. chironomi*	*Diptera*	*Chironomidae*	China; Sweden, and all of Europe	[17,19,26,27,28]
*E. cicadellis*	*Hemiptera*	*Cicadellidae*	Switzerland	[20]
*~E. conica ** (6)	*Diptera*, *Trichoptera*	*Chaoboridae*, *Chironomidae*, *Culicidae*, *Psychodidae*, *Simuliidae*, *Tipulidae*	Asia, incl. Israel; Australia; Europe: Poland, RF, Spain, Switzerland, UK, Ukraine; USA	[17,20,21,23,25,29,30,31,32]
*~E. curvispora ** (3)	*&Diptera*	*Chironomidae*, *Culicidae*, *Psychodidae*, *Simuliidae*	China; Europe: Belarus, Estonia, Poland, RF, Switzerland, Ukraine; Israel; NA	[17,20,21,23,28,30,33,34,35]
*E. delpiniana*	*Diptera*	*Muscidae*	Italy	[17]
*E. fluvialis*	*Diptera*	*Nematocera*	Switzerland	[20]
*E. gigantea*	*Hemiptera*	*Aphrophoridae*	China	[36,37]
*~E. gracilis*	*Diptera*	Minute gnats	Switzerland; Eastern USA	[17,38]
*~E. henrici*	*Diptera*	*Culicidae*	France; Israel	[39]
*E. jaczewskii*	*Coleoptera*	*Carabidae*	Ukraine	[23]
*E. nebriae*	*Coleoptera*	*Carabidae*	Denmark, Germany	[40]
*~E. ovispora ** (2)	*Diptera*	*Calliphoridae*, *Lonchaeidae*, *Muscidae*, *Psychodidae*, *Sarcophagidae*, *Syrphidae*, *Tipulidae*	Asia: Israel, China, RF; Europe: Austria, Poland, RF, Sakartvelo (former Georgia), Sweden, Switzerland, Ukraine; NA	[17,18,19,20,21,23,30,31,37,41]
*~E. plecopteri*	*Plecoptera*	*Nemouridae*	Europe: Spain, Switzerland, UK; NZ	[17,20,25]
*~E. rhizospora* (8)	*Diptera*, *Trichoptera*	*Hydropsychidae*, *Phryganeidae*	China; Europe: Spain, Sweden, Switzerland, UK; USA	[10,17,20,21,32,42]
*~E. sepulchralis* (1)	*Diptera*, *Trichoptera*	*Anthomyzidae*, *Rhagionidae*, *Syrphidae*, *Tipulidae*	Europe: Poland, Ukraine; USA	[10,17,23]
*~E. thurgoviensis*	*Diptera*	*Psychodidae*	Switzerland	[43]
*E. tumefacta*	*Diptera*	*Muscidae*	Switzerland	[20]
*~E. variabilis*	*Diptera*	*Psychodidae*	Europe: Poland, Ukraine, Spain, Sweden, Switzerland; NA	[10,17,20,21,23,25,32]
*Furia americana* (5)	*Diptera*	*Calliphoridae*, *Muscidae*, *Sarcophagidae*	Brazil; Europe: Italy, Switzerland, UK; USA	[10,17,20,20,21,28,32,34]
*F. creatonoti*	*Lepidoptera*	*Erebidae*	China, Sri Lanka, Taiwan	[17,24,28]
*F. ellisiana*	*Dermaptera*	*Forficulidae*	Europe: Poland, Switzerland, UK	[17]
*F. fujiana*	*Lepidoptera*	*Erebidae*	China	[28,44]
*F. fumimontana*	*Diptera*	suborder *Brachycera*	NA; Poland	[3,17]
*F. gastropachae* (13)	*Lepidoptera*	*Lasiocampidae*, *Noctuidae*	Brazil; China; Europe: Ukraine Spain; NA: Canada, USA	[17,23,25,28,45,46,47]
*F. ithacensis* (4)	*Diptera*	*Empididae*, *Rhagionidae*, *Sciaridae*	China; Europe: Poland, Spain; USA	[17,25,28,48]
*~F. montana*	*Diptera*	*Chironomidae*	UK; USA	[10,17,20,21,32,34]
*F. neopyralidarum* (1)	*Lepidoptera*	*Erebidae*, *Pyralidae*, *Tortricidae*	Israel, Japan	[17,49,50]
*F. pieris* (1)	*Lepidoptera*	*Pieridae*, *Zygaenidae*	China; NA, incl. USA	[17,49,50]
*F. shandongensis*	*Dermaptera*	*Forficulidae*	China	[28,51]
*F. triangularis*	*Hemiptera*	*Psyllidae*	China, Philippines	[28,52,53]
*F. virescens* (4)	*Lepidoptera*	*Noctuidae*	Asia: China, Turkmenistan; Europe: Czechia, Finland, Germany, Poland, RF, Spain, Switzerland, UK, Ukraine; NA	[10,17,20,21,25,31,32,33,34,54,55]
*F. vomitoriae*	*Diptera*	*Calliphoridae*, *Stratiomyiidae*, *Syrphidae*	Europe: Austria, Czechia, Poland, RF; Mexico	[17,41,56]
*F. zabri*	*Coleoptera*	*Carabidae*	Europe: Czechia, Ukraine; Uzbekistan	[17,23]
*Orthomyces aleyrodis*	*Hemiptera*	*Aleyrodidae*	USA, Philippines	[4,57]
*Pandora aleurodis*	*Hemiptera*	*Aleyrodidae*	Romania	[17]
*P. batallata*	*Entognatha*, *Symphypleona*	*Sminthuridae*	Germany	[58]
*~P. bibionis*	*Diptera*	*Bibionidae*, *Sciaridae*	China; Switzerland	[28,43,51]
*P. blunckii* (34)	*Hymenoptera*, *Lepidoptera*	*Plutellidae*, *Tenthredinidae*, *Tortricidae*	Asia: Israel, China, Japan, Philippines; Australia; Europe (not reported in Poland); Mexico	[17,20,21,28,42,59,60,61,62,63,64]
*P. borea*	*Diptera*	*Calliphoridae*, *Muscidae*, *Sarcophagidae*	China	[28,65]
*P. brahminae*	*Coleoptera*	*Scarabaeidae*	Bharat, China	[17,28]
*P. bullata*	*Diptera*	*Calliphoridae*, *Sarcophagidae*	Brazil; Australia; Europe: Spain, Switzerland, UK; Iran; NA: USA, Canada; SA	[17,20,20,21,25,66,67,68,69,70]
*P. cacopsyllae* *	*Hemiptera*	*Psyllidae*	Denmark	[71]
*P. calliphorae*	*Diptera*	*Anthomyiidae*	China; France	[28,72]
*~P. chironomid*	*Diptera*	*Chironomidae*	China	[26]
*P. cicadellis*	*Homoptera*	*Cicadellidae*	China	[28,52]
*P. dacnusae*	*Hymenoptera*	*Braconidae*	Poland	[17]
*P. delphacis* (64)	*Hemiptera*	*Delphacidae*	Asia: Bharat, China, East Asia, Indonesia, Japan, Philippines; SA: Brazil, Argentina; Switzerland; USA	[17,28,43,63,64,73,74,75,76,77]
*~P. dipterigena* (8)	*Diptera*	*Calliphoridae*, *Muscidae*, *Mycetophilidae*, *Psychodidae*, *Rhagionidae*, *Sciaridae*, *Syrphidae*, *Tachinidae*, *Tipulidae*	Asia: Bharat, China, Indonesia, Iran, Israel; Brazil; Europe: Austria, Poland, RF, Sakartvelo, Spain, Sweden, Switzerland, UK, Ukraine; NA: Mexico, USA	[10,17,18,19,23,25,28,31,32,34,41,42,69,78,79,80,81,82,83]
*P. echinospora*	*Diptera*, *Hemiptera*	*Aphididae*, *Formicidae*, *Lauxaniidae*	Asia: China, Israel; Europe: Austria, Poland, Sakartvelo, Spain, Sweden, Switzerland, UK, Ukraine; NA: Costa Rica, USA	[10,17,18,20,23,25,28,30,34,41,59,68,84]
*P. formicae*	*Hymenoptera*	*Formicidae*	Bharat; Europe, incl. Denmark	[17,85,86]
*P. gammae* *	*&Lepidoptera*	*Erebidae*, *Noctuidae*	Asia: China, Israel, Turkmenistan; Australia; Europe: Poland, RF, Slovakia, Switzerland, Ukraine; NA, incl. Mexico; SA: Argentina, Brazil	[17,20,21,28,31,33,56,67,80,87,88,89,90,91,92,93]
*~P. gloeospora*	*Diptera*	*Mycetophilidae*, *Psychodidae*, *Sciaridae*	China; Europe: France, Ukraine; USA	[15,17,23,24,28,94]
*P. guangdongensis*	*Hemiptera*	*Miridae*	China	[95]
*P. heteropterae* (1)	*Hemiptera*	*Miridae*	NA, incl. USA; Poland	[17,96]
*P. kondoiensis* * (5)	*Hemiptera*	*Aphididae*	Australia; China	[17,28,65,97]
*P. lipae*	*Coleoptera*	*Cantharidae*	Denmark, France, Poland, Switzerland	[17,21]
*~P. longissimi*	*Diptera*	*Limoniidae*	Switzerland	[20]
*P. minutispora*	*Hemiptera*	*Miridae*	Czechia, Switzerland	[17,20]
*P. muscivora*	*Diptera*	*Calliphoridae*, *Drosophilidae*, *Muscidae*, *Tachinidae*	Canada; Europe: Poland, UK, Ukraine	[17,23,34]
*P. myrmecophaga*	*Hymenoptera*	*Formicidae*	Brazil; Europe: Czechia, Germany, Poland, Sweden, Switzerland, Ukraine, former Yugoslavia; Philippines	[17,20,21,23,64]
*P. neoaphidis* * (TYPE, 173)	*Hemiptera*	*Aphididae*	Worldwide, less frequent in tropics: Africa: Egypt, Tunisia, South Africa; Asia: Bharat, China, Iran, Israel, Japan, Korea, Nepal, Philippines, Taiwan; Australia; Europe: Austria, Bosnia and Herzegovina, Denmark, Finland, France, Iceland, Latvia, Poland, Portugal, RF, Serbia, Slovakia, Spain, Switzerland, UK, Ukraine; NA: Canada, Mexico, USA; NZ; SA: Argentina, Brazil, Chile, Uruguay	[17,20,21,22,23,28,30,31,41,51,54,56,59,62,64,69,78,79,83,97,98,99,100,101,102,103,104,105,106,107,108,109,110,111,112,113,114,115,116,117,118,119]
*P. nouryi* * (5)	*Hemiptera*, *&Psocodea*	*Aphididae*, *Pseudocaeciliidae*	Argentina; Asia: China, Israel; Australia; Europe: Central, Northern, and Western, incl. Slovakia; NA	[17,28,51,59,79,101,110,111,113,120,121]
*P. phalangicida*	*Mesostigmata*, *&Opiliones (Arachnida)*	*Parasitidae*	Poland, Sweden, UK	[17,34]
*P. philonthi*	*Coleoptera*	*Staphylinidae*	Denmark, Poland, Switzerland	[17,20,122]
*P. phyllobii*	*Coleoptera*	*Curculionidae*	Poland	[17]
*P. polonae-majoris*	*Hemiptera*	*Cicadellidae*, *Jassidae*	Poland, Ukraine	[17,23]
*P. psocopterae*	*Psocodea*	*Prionoglarididae*	France	[17,20,21]
*~P. sciarae*	*Diptera*	*Sciaridae*	Europe: Austria, Denmark, Switzerland, Ukraine; NZ; USA	[17,20,23,32,41]
*P. shaanxiensis*	*Diptera*	*Calliphoridae*	China	[26,28]
*P. sylvestris* sp. nov.	*Lepidoptera*	*Erebidae*	USA	Hajek and Gryganskyi, in press
*P. terrestris*	*Hemiptera*	*Aphididae*	Ukraine	[120,123]
*P. uroleuconii*	*Hemiptera*	*Aphididae*	Slovakia	[124]
*Strongwellsea acerosa*	*Diptera*	*Muscidae*	Denmark	[125]
*S. castrans* (2)	*Diptera*	*Anthomyiidae*	China; Europe: Czechia, Denmark, Switzerland, UK; USA	[28,60,126,127]
*S. crypta*	*Diptera*	*Anthomyiidae*	Denmark	[128]
*S. gefion*	*Diptera*	*Muscidae*	Denmark	[129]
*S. magna*	*Diptera*	*Fanniidae*	China; USA; Denmark	[37,130]
*S. pratensis*	*Diptera*	*Muscidae*	Switzerland	[20]
*S. selandia*	*Diptera*	*Muscidae*	Denmark	[129]
*S. tigrinae*	*Diptera*	*Muscidae*	Denmark	[125]
*Strongwellsea* sp. nov.	*Diptera*	*Calliphoridae*	Europe	[131]
*Strongwellsea* sp. nov.	*Diptera*	*Sarcophagidae*	Europe	Eilenberg, unpublished
*Strongwellsea* sp. nov.	*Diptera*	*Scatophagidae*	Europe	[132]
*Strongwellsea* sp. nov.	*Diptera*	*Anthomyiidae*	Europe	Eilenberg et al., unpublished
*Zoophthora anglica ** (5)	*Coleoptera*	*Elateridae*	Denmark, France, Poland, Romania, Switzerland, UK, Ukraine	[17,23,133]
*Z. anhuiensis*	*Hemiptera*	*Aphididae*	China	[17,28,62,134]
*Z. aphidis ** (1)	*Hemiptera*, *Lepidoptera*	*Aphididae*, *Cicadellidae*, *Delphacidae*, *Erebidae*	Asia: China, Philippines, Taiwan; Europe: Armenia, Belarus, Estonia, Lithuania, Moldova, RF, Sweden, UK, Ukraine; NA: Canada, Puerto Rico, USA	[19,23,31,32,34,57,106,112,135,136]
*Z. aphrophorae*	*Hemiptera*	*Aphrophoridae*, *Cicadellidae*, *Miridae*, *Psyllidae*	UK	[34]
*Z. arginis*	*Hymenoptera*	*Argidae*	Germany, Poland	[17,29]
*Z. athaliae*	*Hymenoptera*	*Tenthredinidae*	China; Switzerland	[17,20,28,65]
*Z. autumnalis*	*Diptera*	*Dryomyzidae*	Poland	[17]
*Z. bialovienzensis **	*&Lepidoptera*	*Geometridae*, *Pyralidae*	Poland, Ukraine	[17,23]
*Z. brevispora*	*Lepidoptera*	*Geometridae*	Poland	[17]
*Z. canadensis*	*Hemiptera*, *Lepidoptera*	*Aphididae*, *Geometridae*	China; NA	[137,138]
*Z. crassispora*	*Lepidoptera*	*Tortricidae*	Poland	[17]
*Z. crassitunicata*	*Coleoptera*	*Cantharidae*	Austria, Switzerland	[17,20,41]
*Z. elateridiphaga*	*Coleoptera*	*Elateridae*	Switzerland	[21]
*Z. erinacea*	*Hemiptera*	*Aphididae*	Israel; Slovakia	[17,59,79,120]
*Z. falcata*	*Hymenoptera*	*Formicidae*	Poland	[17]
*Z. forficulae*	*Dermaptera*	*Forficulidae*	Europe: Poland, Switzerland, UK; NA	[17,32,34,43]
*Z. geometralis*	*&Lepidoptera*	*Geometridae*, *Yponomeutidae*	Europe: Austria, Sweden, Ukraine; NA	[10,17,18,23,41]
*Z. giardia*	*Orthoptera*	*Tettigoniidae*	France, Germany, Poland	[17]
*Z. humberi*	*Diptera*	*Tipulidae*	Chile	[17,139]
*Z. ichneumonis **	*&Hymenoptera*	*Ichneumonidae*	Poland, Switzerland, Ukraine	[17,20,23]
*Z. independentia*	*Diptera*	*Tipulidae*	USA	[140]
*Z. lanceolata ** (3)	*Diptera*, nematodes	*Drosophilidae*, *Empididae*	Europe: France, Poland, Spain, Switzerland, Ukraine; Israel	[17,23,25,30,141]
*Z. larvivora*	*Coleoptera*	*Cantharidae*	Poland	[17]
*Z. miridis*	*Hemiptera*	*Miridae*	Poland, Spain, Switzerland	[17,25,141]
*~Z. nematoceris*	*Diptera*	*Bibionidae*, *Sciaridae*	Poland, Spain, Switzerland	[17,25,141]
*Z. obtusa*	*Diptera*	*Brachycerous*, *Calyptrate*	Poland, Switzerland	[17,43]
*Z. occidentalis* (7)	*Hemiptera*	*Aphididae*	Asia, incl. China; Europe: Poland, Slovakia, Spain, Switzerland, UK; NA; SA, incl. Chile	[17,20,25,32,34,78,79,103,104,139,142,143]
*Z. opomyzae*	*Diptera*	*Opomyzidae*	Austria, Germany, Poland	[17,29,41]
*Z. orientalis*	*Hemiptera*	*Aphididae*	Israel	[30]
*Z. pentatomis*	*Hemiptera*	*Pentatomidae*	China	[28,51,52,137]
*Z. petchii*	*Hemiptera*	*Aphrophoridae*, *Cercopidae*, *Cicadellidae*, *Delphacidae*	Asia: China, Israel; Europe: Austria, Switzerland	[17,20,29,30,74]
*Z. phalloides* (2)	*Hemiptera*	*Aphididae*	Argentina; Asia: Israel, Korea; Europe: Germany, Poland, Slovakia, Switzerland, UK; Australia NZ; NA, incl. Mexico	[17,20,21,29,30,78,79,83,99,107,144,145,146]
*Z. phytonomi **	*&Coleoptera*	*Curculionidae*	Asia: Israel, Uzbekistan; Australia; Europe: Poland, Romania, Ukraine; USA	[17,23,30,147]
*Z. porteri*	*Diptera*	*Tipulidae*	Ukraine; USA	[23,140]
*Z. psyllae*	*Hemiptera*	*Psyllidae*, *Triozidae*	Poland, Spain, Switzerland	[17,20,25]
*Z. radicans ** (305)	*Diptera*, *Coleoptera*, *Hemiptera*, *Homoptera*, *Hymenoptera*, *&Lepidoptera*, *Plecoptera*	*Agromyzidae*, *Aphididae*, *Aphrophoridae*, *Aleurodidae*, *Argidae*, *Chironomidae*, *Chrysomelidae*, *Cicadellidae*, *Crambidae*, *Delphacidae*, *Geometridae*, *Miridae*, *Muscidae*, *Nemouridae*, *Pentatomidae*, *Pieridae*, *Plutellidae*, *Psyllidae*, *Triozidae*, *Thaumastocoridae*, *Tortricidae*	Africa: South Africa, Tchad; Asia: China, Indonesia, Israel, Japan, Korea, Kyrgyzstan, Malaysia, Philippines; Australia, NZ; Europe: Belarus, Denmark, Estonia, France, Moldova, Poland, RF, Sakartvelo, Serbia, Slovakia, Sweden, Switzerland, UK, Ukraine, former Yugoslavia; NA: Canada, Cuba, Mexico, Puerto Rico, USA; SA: Argentina, Brazil, Uruguay	[17,18,20,21,23,28,30,31,32,34,51,56,61,62,64,78,79,83,92,93,93,95,113,114,116,122,137,146,148,149,150,151,152,153,154,155,156,157,158,159,160,161,162,163,164]
*Z. rhagonycharum*	*Coleoptera*	*Cantharidae*	Europe: Denmark, Poland, Switzerland; NA	[17,20,165]
*Z. suturalis*	*Coleoptera*	*Chrysomelidae*	France, UK	[17]
*Z. tachypori*	*Coleoptera*	*Staphylinidae*	Poland	[17]
*Z. viridis*	*Hemiptera*	*Miridae*	Western Europe, incl. Germany, Switzerland	[17,21]
Total 123 species, in ARSEF—683 specimens of 28 species.	Total 14 orders.	Total 76 families.	Total 57 countries, in ARSEF specimens from 29 countries.	Total 160 records.

Continents and countries placed alphabetically, ~—aquatic or moist habitats, *—cultivable, &—infects other or more stages than adult, in parentheses number of specimens in ARSEF culture collection, NA—North America, RF—Russian Federation, SA—South America, UK—United Kingdom. French Guiana is merged with France in the map (Figure 1), but EFOPSZ records do not come from French Guiana but rather from France in Europe.

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
