# Peer review of "Potential for Use of Species in the Subfamily Erynioideae for Biological Control and Biotechnology"

_microorganisms, 2024, doi:10.3390/microorganisms12010168_

Round 1

Reviewer 1 Report

Comments and Suggestions for Authors

Dear Authors,

In my opinion, the authors have done an excellent review of the main aspects of the life cycle of some genera in the Erynioideae. They have carried out an extensive bibliographic review that has allowed them to generate a map of the global distribution of these entomopathogens and create phylogenetic trees to establish the relationships between the genera and species recovered. I consider very interesting and necessary the information provided about the advances in the search for species and technologies that will allow this group of fungi to be used in the future as biological control agents for agricultural pests or insects of health interest. Based on the above, I think this manuscript should be considered for publication in its current form. The authors should only correct minor errors that I detail below.

Lines 65 and 382: Please write in vitro in italics.

Line 382: Please write in vivo in italics.

Line 396: Replace fro by of.

Author Response

Reviewer I.

In my opinion, the authors have done an excellent review of the main aspects of the life cycle of some genera in the Erynioideae. They have carried out an extensive bibliographic review that has allowed them to generate a map of the global distribution of these entomopathogens and create phylogenetic trees to establish the relationships between the genera and species recovered. I consider very interesting and necessary the information provided about the advances in the search for species and technologies that will allow this group of fungi to be used in the future as biological control agents for agricultural pests or insects of health interest. Based on the above, I think this manuscript should be considered for publication in its current form. The authors should only correct minor errors that I detail below.

Dear Reviewer I. We thank you for your very positive evaluation of our manuscript. We made all the necessary changes you suggested.

Lines 65 and 382: Please write in vitro in italics.

Line 382: Please write in vivo in italics.

Changed to italics, also through the text.

Line 396: Replace fro with of.

Replaced.

Reviewer 2 Report

Comments and Suggestions for Authors

The manuscript, " Potential use of species in the Erynioideae for biological control and biotechnology " provides a comprehensive overview of the current state of research on entomopathogenic fungi within the Erynioideae subfamily, including Erynia, Furia, Pandora, Strongwellsea, and Zoophthora orders. The authors delve into the distribution, biology, and genetics of these fungi, with practical applications for pest control discussed in the manuscript. The information presented is valuable and could be published after some enhancements.

Section 3.3: I recommend including a more detailed scheme illustrating the interactions between fungi the Erynioideae subfamily and hosts. This addition could help readers to explain specific points of the fungi's biology.

Photography: Consider incorporating photographs, possibly scanning electron microscopy (SEM) images, of the fungi (Erynia, Furia, Pandora, Strongwellsea, and Zoophthora). Visual aids can enhance reader comprehension.

Lines 230-233: It appears that these lines might be a repetition of lines 142-145. Please verify and eliminate any redundancy.

Section 3.5: It would be beneficial to provide a concise summary of current approaches to genetically identifying fungal species within Erynioideae. This could include specific genomic regions or primers used for identification purposes.

Lines 291-295: The statement about entomopathogenic fungi producing a wide range of toxins is intriguing. However, additional examples, especially those related to metabolites used in plant protection, would strengthen the section. Additionally, the authors should offer more detailed information about the fungi covered in the review (Erynia, Furia, Pandora, Strongwellsea, and Zoophthora), as this could be crucial for readers.

Overall, with these suggested improvements, the manuscript has the potential to be an even more valuable resource for the journal's audience.

Author Response

Reviewer II.

The manuscript, " Potential use of species in the Erynioideae for biological control and biotechnology " provides a comprehensive overview of the current state of research on entomopathogenic fungi within the Erynioideae subfamily, including Erynia, Furia, Pandora, Strongwellsea, and Zoophthora orders. The authors delve into the distribution, biology, and genetics of these fungi, with practical applications for pest control discussed in the manuscript. The information presented is valuable and could be published after some enhancements.

Dear Reviewer, thank you for your positive response and valuable suggestions. We added one more genus, which is rarely mentioned as a representative of this fungal group but belongs to it according to the available morphological descriptions.

Section 3.3: I recommend including a more detailed scheme illustrating the interactions between fungi the Erynioideae subfamily and hosts. This addition could help readers to explain specific points of the fungi's biology.

Unfortunately, we were unable to create the scheme illustrating the interaction between the fungi and their hosts, mainly because of luck of drawing capacities in our research groups. However, we added a plentiful plate of photographs illustrating the most important aspects of EFOPSZ biology, which will improve the reader’s understanding of the matters discussed in our manuscript.

Photography: Consider incorporating photographs, possibly scanning electron microscopy (SEM) images, of the fungi (Erynia, Furia, Pandora, Strongwellsea, and Zoophthora). Visual aids can enhance reader comprehension.

We added the plate with photographs, presenting various EFOPSZ fungi and different aspects of their biology, showing infected insects and fungal microstructures.

Lines 230-233: It appears that these lines might be a repetition of lines 142-145. Please verify and eliminate any redundancy.

Redundancy removed.

Section 3.5: It would be beneficial to provide a concise summary of current approaches to genetically identifying fungal species within Erynioideae. This could include specific genomic regions or primers used for identification purposes.

We added another paragraph to this section presenting the current situation with the availability of molecular data for the EFOPSZ fungi.

Lines 291-295: The statement about entomopathogenic fungi producing a wide range of toxins is intriguing. However, additional examples, especially those related to metabolites used in plant protection, would strengthen the section. Additionally, the authors should offer more detailed information about the fungi covered in the review (Erynia, Furia, Pandora, Strongwellsea, and Zoophthora), as this could be crucial for readers.

We added some info about toxins and secondary metabolites, which definitely improved the genome part of our manuscript.

Overall, with these suggested improvements, the manuscript has the potential to be an even more valuable resource for the journal's audience.

Thank you again for your very encouraging evaluation of our work.

Reviewer 3 Report

Comments and Suggestions for Authors

General information:

This is a very interesting manuscript presenting a thorough literature review on the biocontrol potential of the Erynioideae subfamily. I appreciate the semi-systemic approach providing both a thorough literature search and an expert narrative. I only have a few minor comments for this manuscript. The language of the abstract could be improved as it is not the easiest to follow. I would suggest using simpler language and shorter sentences for this part since it is the first part to be read so the reader can easily understand the topic of your manuscript. In this review you mention the importance of host specificity and geographic distribution, therefore I lack a graphical representation of the comparison of Erynioideae families in that regard (more details below). I understand that the comparison of such information might be biased by different sampling across different regions, but I still believe it can give some indications. From your analysis, it is clear that the tested taxa are more studied in countries like Poland, Switzerland, and the UK, and there could be also bias including sampling sites as most insects will be collected where they can be spotted most easily. Could you propose a compensation method for this bias? Overall I consider this manuscript acceptable for publication in microorganisms after minor revisions and will be interested to read it in its final form.

In-text comments:

Line 2: do not divide words in title. Consider changing the title: e.g. “Potential use of Erynioideae species in the for biological control and biotechnology” or “Potential use of species from Erynioideae subfamily for biological control and biotechnology”

Line 30: than any other fungi

Line 30: this sentence is unclear: “Given their potential to attack arthropods and in the fungal evolutionary tree…”

Line 88: Please provide the date range of the literature search and used keywords.

Line 151: provide full name

Line 165: Consider using a bar plot or table as more easy to follow for such data

Consider adding a similar figure concerning genera distribution and host specificity.

Comments on the Quality of English Language

The abstract could be slightly improved. Please try to use simpler language and shorten sentences.

Author Response

Reviewer III.

This is a very interesting manuscript presenting a thorough literature review on the biocontrol potential of the Erynioideae subfamily. I appreciate the semi-systemic approach providing both a thorough literature search and an expert narrative. I only have a few minor comments for this manuscript. The language of the abstract could be improved as it is not the easiest to follow. I would suggest using simpler language and shorter sentences for this part since it is the first part to be read so the reader can easily understand the topic of your manuscript. In this review you mention the importance of host specificity and geographic distribution, therefore I lack a graphical representation of the comparison of Erynioideae families in that regard (more details below). I understand that the comparison of such information might be biased by different sampling across different regions, but I still believe it can give some indications. From your analysis, it is clear that the tested taxa are more studied in countries like Poland, Switzerland, and the UK, and there could be also bias including sampling sites as most insects will be collected where they can be spotted most easily. Could you propose a compensation method for this bias? Overall I consider this manuscript acceptable for publication in microorganisms after minor revisions and will be interested to read it in its final form.

Dear Reviewer, thank you for your very positive evaluation of our manuscript and valuable suggestions. Regarding the largely lacking sampling of this group, we suggest that it is useful to provide such a map and highlight the best-sampled regions. This will give the idea of how many species can be in each region of our planet with suitable climates. Our responses to the comments in the text below, are underlined.

In-text comments:

Line 2: do not divide words in title. Consider changing the title: e.g. “Potential use of Erynioideae species in the for biological control and biotechnology” or “Potential use of species from Erynioideae subfamily for biological control and biotechnology”

We changed the title to “Potential for Use of Species in the Subfamily Erynioideae for Biological Control and Biotechnology.”

Line 30: than any other fungi

We changed this phrase to “than many other fungi”

Line 30: this sentence is unclear: “Given their potential to attack arthropods and in the fungal evolutionary tree…”

We changed this sentence to: “Given their potential to attack arthropods and their position in the fungal evolutionary tree, we discuss which species might be adopted for biological control purposes or biotechnological innovations.”

Line 88: Please provide the date range of the literature search and used keywords.

We added this information to the Material & Methods section.

Line 151: provide full name.

  1. changed to Pandora.

Line 165: Consider using a bar plot or table as more easy to follow for such data.

We changed the pie diagram to plot one, also revised the number of taxa, and made correspondent corrections.

Consider adding a similar figure concerning genera distribution and host specificity.

We decided not to add too many figures. Figure 3 is the most general one, genera or host distribution will be more crowded and not easy to compile. We believe that Figure 3 gives a sufficient hint to the host specificity of the whole EFOPSZ group.

The abstract could be slightly improved. Please try to use simpler language and shorten sentences.

We shortened the sentences in the Abstract and improved the language, so that is easier to understand and smoother. Thank you again for your very encouraging review.
